# Novel 1-(2-Aryl-2-adamantyl)piperazine Derivatives Exhibit In Vitro Anticancer Activity Across Various Human Cancer Cell Lines, with Selective Efficacy Against Melanoma

**DOI:** 10.3390/medicina61101731

**Published:** 2025-09-23

**Authors:** Irida Papapostolou, Evangelia Sereti, Stavroula Chatira, Nikos Sakellaridis, George Fytas, Grigoris Zoidis, Konstantinos Dimas

**Affiliations:** 1Department of Biochemistry and Molecular Medicine, Faculty of Medicine, University of Bern, 3012 Bern, Switzerland; 2Department of Pharmacology, Faculty of Medicine, Health Sciences School, University of Thessaly, 41500 Larissa, Greece; evangelia.sereti@med.lu.se (E.S.); stavroula.chatira@gmail.com (S.C.); nsakella@uth.gr (N.S.); 3Department of Translational Medicine, Faculty of Medicine, Lund University, 21428 Malmö, Sweden; 4Department of Pharmaceutical Chemistry, National and Kapodistrian University of Athens, 15772 Athens, Greece; gfytas@pharm.uoa.gr (G.F.); zoidis@pharm.uoa.gr (G.Z.)

**Keywords:** melanoma, 1-(2-aryl-2-adamantyl)piperazine derivatives, sigma receptors, sigma ligands, autophagy, apoptosis

## Abstract

*Background and Objectives*: Cutaneous melanoma (CM) is widely regarded as the most aggressive form of skin cancer worldwide, showing a rising global incidence. It develops from the uncontrolled transformation of pigment-producing melanocytes. The aim of this study is to characterize the cytotoxic and anti-proliferative properties of two 1-(2-aryl-adamantyl)piperazine derivatives, **6** and **7**, with a specific emphasis on their impact on melanoma cells. Both compounds are synthesized based on the adamantane core structure which increases drug-like properties of the lead compound phencyclidine I, without increasing toxicity. *Materials and Methods*: This study describes concentration-dependent effects on cell viability and clonogenicity. *Results:* SRB assays, clonogenic (long-term) assays, and scratch assays reveal a significant anticancer activity of these two agents at low μΜ levels with a selective activity against melanoma cells. Furthermore, Western blot experiments indicate that both **6** and **7** induce LC3 accumulation, procaspase 3 decrease, and PARP cleavage, suggesting the implication of multiple death pathways in their anticancer mechanism of action. *Conclusions:* This study sheds light on the in vitro anticancer potential of two novel 1-(2-aryl-2-adamantyl)piperazine derivatives. It highlights their differential activity against melanoma and emphasizes their potential as lead candidates for further therapeutic exploration.

## 1. Introduction

In 2020, the World Health Organization (WHO) reported 19.3 million global cancer cases and approximately 10 million cancer-related deaths. Among these, melanoma caused an estimated 325,000 new cases and 57,000 deaths globally [1]. Within the European Union, melanoma is among the most frequently diagnosed types of cancer, in both sexes, according to recent data by the European Commission [2].

Melanoma develops when melanocytes, the cells responsible for skin pigmentation, proliferate uncontrollably. When detected at an early stage, melanoma has a favorable prognosis, with a five-year survival rate of around 92% [3,4]. However, early-stage diagnosis can be clinically challenging, due to the difficulty in identifying malignant versus benign lesions. Both environmental and genetic parameters can result in the development of melanoma. Ultraviolet (UV) light is the most prominent environmental risk factor for melanoma development [5]. At the genetic level, gene mutations play a significant role in the development of melanoma, with BRAF V600 and NRAS being the most common. However, not all melanoma tumors carry these mutations, highlighting the disease’s complexity and the need to further study its genetic background [6]. Advances in melanoma treatment include targeted therapies and immune checkpoint inhibitors, though their benefits are limited by resistance, toxicity, and high costs. Interestingly, many recent studies (summarized in an informative review by Slominski et al. [7]) report in vivo data, as well as data from clinical trials, which suggest that vitamin D and its derivatives could play a role in improving the response to immunotherapy and reducing the incidence of melanoma. It is of great interest that integrating vitamin D signaling with computational biology and artificial intelligence (AI)-driven strategies could help to overcome resistance and optimize personalized therapies.

In a study by Ferle-Vidović and colleagues, the in vitro anti-proliferative activity of phencyclidine I (1-(1-phenylcyclohexyl)piperidine) (Figure 1) and its rigid analogues bearing an adamantyl group in place of the cyclohexyl ring [8] were reported. The adamantane group has numerous applications in drug development and medicinal chemistry and is found in several bioactive compounds with activity against viruses [9], bacteria [10], and parasites (e.g., Trypanosoma) [11], as well as for the treatment of human diseases such as cancer and Parkinson’s [12,13,14]. Adamantane derivatives have been previously reported to show potent anticancer activities in vitro and in vivo against a variety of human cancer cell lines and against human-to-mouse xenografts [15,16,17]. C1-substituded adamantanes were reported to show in vitro activity against a panel of human cancer cell lines (including the melanoma cells LOX-IMVI) and in vivo activity against pancreatic cancer xenografts [15], while adamantane phenylalkylamines with σ-receptor binding affinity have been reported to show potent antimetastatic activity against the human PC3 prostate cancer xenograft together with analgesic activities [17]. More recently, as part of our ongoing efforts to develop adamantane derivatives with anticancer activity, we have designed and synthesized a number of novel 1-(2-aryl-2-adamantyl)piperazine derivatives [18]. In these analogues, the cyclohexane ring of phencyclidine I was replaced by the more lipophilic and drug-like adamantane ring, while the piperidine moiety was replaced by piperazine, offering an additional amino group (NH) that might act as a hydrogen-bonding or electrostatic interaction site [18]. Among them, the 1-(2-phenyl-2-adamantyl)piperazine **6** and the 1-[2-(4-fluorophenyl)-2-adamantyl] piperazine **7** (Figure 1) show the most promising anti-proliferative activity across four established human cancer cell lines, HeLa, MDA-MB-231, MIA PaCa-2, and NCI-H1975. Additionally, the presence of the fluorine group at the benzene ring further enhances the antitumor activity. Affinity analyses suggest that compounds **6** and **7** may serve as potential sigma receptors’ inhibitors, although their classification as antagonists or agonists of the sigma receptors remains to be determined [18].

Building on our previous findings [18], the present study aimed to characterize the anticancer activities of adamantyl-piperazine derivatives further and to elucidate aspects of their underlying mechanisms of action. Our results highlight the in vitro anticancer potential of these compounds, which displayed selective activity against melanoma under the experimental conditions examined. The data also suggest a pleiotropic mechanism of action, likely involving both apoptosis and autophagy.

## 2. Materials and Methods

### 2.1. Cell Lines and Cell Culture

The cell lines used in this study are listed in Table 1. All cell lines were purchased from the National Cancer Institute (NCI) (Bethesda, MD, USA), with the exception of AcPC-1 and BxPC-3, which were purchased from ATCC (Washington, DC, USA). Cells were cultured at 37 °C and 5% CO_2_ in RPMI 1640 (Gibco, Waltham, MA, USA, Cat. No. 31870-025) supplemented with 5% fetal bovine serum (FBS) (Biowest, Nuaillé, France, Cat. No. 51810-500), 1% penicillin-streptomycin (Gibco, Waltham, MA, USA, Cat. No. 15140-122), and 1% L-glutamine (Gibco, Waltham, MA, USA, Cat. No. 25030-024).

### 2.2. Synthesis of 1-(2-Aryl-2-adamantyl)piperazine Derivatives

The 1-(2-aryl-2-adamantyl)piperazine derivatives were synthesized as previously published [19]. Two-dimensional NMR experiments (HMQC and COSY) were performed for the elucidation of the structures of the newly synthesized compounds. Analytical thin-layer chromatography (TLC) was conducted on Merck silica gel 60 F_254_ (Merck KGaA, Darmstadt, Germany) precoated plates (0.2 mm layer thickness), with the spots visualized by iodine vapors under UV light. Column chromatography purification was carried out on silica gel 60 (70–230 mesh). Elemental analyses (C, H, N) were performed by the Service Central de Microanalyse at CNRS (France) and were within ±0.4% of the theoretical values. Yields refer to chromatographically and spectroscopically homogeneous materials. Compound purities were determined by elemental analysis. The results obtained correspond to >95% purity. The commercial reagents were purchased from Alfa Aesar (Haverhill, MA, USA), Sigma Aldrich (St. Louis, MO, USA), and Merck (Darmstadt, Germany), and were used without further purification. Organic solvents were of the highest purity, and when necessary, dried according to standard procedures. CLogP and PKa values for 6 and 7 were calculated using ChemDraw (v21 July 2021, Revvity Signals Software, Inc., Waltham, MA, USA).

### 2.3. Cell Viability Assays

#### 2.3.1. Sulforhodamine B (SRB)

To study the effect of the compounds on cancer cells’ viability, the SRB colorimetric assay was performed. Cells were seeded in 96-well plates at optimized densities in 100 μL of complete growth medium, allowed to grow for 24 h, and then treated with different concentrations (0.1, 1, 10, and 100 μΜ) of compounds **6** and **7**. For AcPC-1 and BxPC-3 the plating densities were 5000 each, while for the rest, as has been reported in previous works [19]. Prior to the addition of the compounds, 6 wells of cells from every experiment were fixed with 50 μL of 50% *v*/*v* trichloroacetic acid (TCA) (Applichem, Darmstadt, Germany, Cat. No. A1431, 1000) for 1 h at 4 °C. Cells were then washed with PBS 1X, air-dried, and stained with 50 μL of 0.4% *w*/*v* SRB dye (Sigma Aldrich, St. Louis, MO, USA, Cat. No. S9012-5G) diluted in 1% *v*/*v* acetic acid (A.A.) for 10 min at room temperature (RT). Excess dye was washed away with 1% A.A., and the cells were air-dried. Bound dye was solubilized in 150 μL Tris Base 10 mM (ChemCruz-Santa Cruz Biotechnology Inc., Dallas, TX, USA, LOT: A0818), and the cells were incubated at 37 °C for 10 min. Absorbance was measured at 540 nm in a microplate reader (Biotek Instruments, Winooski, VT, USA, model ΕΙ-311). The remaining cells were treated with compounds **6** and **7** or vehicle control for 48 h. Cells were then fixed and stained as described above. From the generated dose–response curves, we determined the GI_50_ (50% cell growth inhibition), TGI (total growth inhibition), and LC_50_ (50% cell death) parameters [20]. Growth inhibition of 50% (GI_50_) is defined by the formula (Ti − Tz)/(C − Tz)) × 100 = 50. Total growth inhibition (TGI) is calculated by the formula Ti = Tz. Finally, the LC_50_, indicating a net loss of cells following treatment, is calculated using the formula (Ti − Tz)/Tz) × 100 = −50. In these formulas, Tz represents a measurement (absorbance) of the cell population for each cell line at the time of drug addition (i.e., 24 h after plating the cells), C represents the measurement of the control cells, and Ti represents the growth of cells in the presence of a given concentration of the drug at the end of the 48 h incubation period [20].

#### 2.3.2. Trypan Blue Exclusion

To further evaluate the cell viability effects of 1-(2-aryl-2-adamantyl)piperazine derivatives, the trypan blue exclusion assay was used in MDA-MB-435 cells. One day prior to treatment, the cell culture medium was refreshed for optimal growth conditions. The next day, cells were dissociated with 0.25% trypsin (Gibco, 15090-046), seeded into T75 flasks (ThermoScientific Nunc, Roskilde, Denmark, Cat. No. 156499) at a density of 1 × 10^6^ cells per flask, and allowed to adhere for 24 h. Before adding the compounds, 100 μL of the culture from each flask was collected at time zero (T0). The samples were mixed 1:1 with a 0.5% (*w*/*v*) trypan blue solution (Biochrom, Cambridge, UK, Cat. No. L6323), incubated at room temperature for 2–3 min, and then assessed for cell viability using a hemocytometer under a microscope. Cells were then treated with compounds **6** and **7** at final concentrations of 10 and 20 μΜ for 6, 12, 24, and 48 h. Parallel aliquots of the same cell populations were used for downstream protein analysis by Western blotting. At each subsequent time point, treated cells were similarly harvested, stained with the trypan blue, counted, and used for protein analysis. Time-matched, untreated cells were included as negative controls for all time points. Viability was expressed as the percentage of trypan blue-negative (viable) cells relative to the total cell number. The percentage of trypan blue-positive (non-viable) cells was used for compound-induced cytotoxicity. Images were acquired under a Zeiss axiovert inverted microscope equipped with an AxioCam ERc 5s camera (Antisel A. Selidis Bros S.A., Thessaloniki, Greece).

### 2.4. Electrophoresis and Western Blot Analysis

Cells were lysed in ice-cold 1X RIPA buffer (Cell signaling, Danvers, MA, USA, Cat. No. 9806S) supplemented with 1% protease and phosphatase inhibitor cocktail (Cell signaling, Danvers, MA, USA, Cat. No. 5872). Cells were kept on ice for 20 min, with intermittent vortexing every 5 min, and centrifuged at 1300 rpm for 30 min at 4 °C. The supernatants were collected, and protein concentration was quantified using the Pierce BCA protein assay kit (ThermoScientific, Waltham, MA, USA, Cat. No. 23227). Equal amounts of total protein (25 μg) were resolved on 10 and 12% acrylamide gels and transferred to 0.22 μm pore size polyvinylidene difluoride (PVDF) membranes (Immobilon-PSQ transfer membrane Millipore, Burlington, MA, USA, ISEQ85R). Membranes were blocked in blocking buffer for one hour at RT followed by overnight incubation at 4 °C with the following primary antibodies: b-actin (42 kDa; Santa Cruz, Heidelberg, Germany, sc-8432; dilution 1:3000), procaspase 3 (17 kDa; Cell signaling, #14220; dilution 1:1000), LC3 A/B-I/II (14, 16 kDa; Cell signaling, #12741; dilution 1:1000), PARP (116 and 89 kDa; Cell signaling, #9542; dilution 1:1000), and GβL (37 kDa; Cell signaling, #3274; dilution 1:1000). All primary antibodies were diluted in the manufacturer’s recommended buffer, supplemented with 1% FBS. Membranes were incubated for 1 h at RT with horseradish peroxidase (HRP)-conjugated secondary antibodies: anti-mouse-HRP (Cell signaling, 7076) and anti-rabbit-HRP (Cell signaling, 7074S), both diluted 1:10,000 with the manufacturer’s recommended buffer, supplemented with 1% FBS. Membranes were developed using the Clarity™ Western Enhanced Chemiluminescence (ECL) substrate (Bio-Rad, 170-50G) and detected with Uvitec Cambridge chemiluminescence imaging system, using Alliance software (ver. 16.06) (Uvitec Cambridge, Cambridge, UK).

### 2.5. Clonogenic/Long-Term Assay

The long-term anti-proliferative effects of compounds **6** and **7** on MDA-MB-435 cells were studied with the clonogenic assay. Based on the results from SRB and trypan blue viability assays, final concentrations of 1, 2.5, and 5 μΜ were used for both compounds. Cells were seeded at a density of 400 cells per well in 6-well plates (ThermoScientific, 140685) in 1 mL RPMI 1640. After 24 h of incubation, an additional 1 mL of fresh medium with indicated concentrations of compounds **6** and **7** was added. In the untreated control cells, 1 mL of complete cell culture medium was added. The cells were cultured for 15 days, with the medium (supplemented with the compounds, where applicable) refreshed every 7 days. After 15 days of incubation, colonies were washed with PBS 1X and fixed with 500 μL of 50% TCA for an hour at 4 °C. Plates were then rinsed with ddH_2_O, air-dried, and stained with 1 mL of 0.4% SRB w/v solution for 10 min at RT. The excess dye was washed away with 1% A.A, and plates were air dried. For quantification, 1 mL of Tris Base 10 mM was added per well, and absorbance was measured at 540 nm with a BioTek ELISA microplate reader (Biotek, ΕΙ-311). Colonies were also counted using ImageJ2 (Fiji version 1.54p).

### 2.6. Statistical Analysis

All quantitative data are presented as the mean ± SEM, with *n* ≥ 2 independent experiments. A one-way analysis of variance (ANOVA) with either Dunnett’s, Šidák, or Holm–Šidák post hoc tests was used to determine statistical significance. Differences were considered significant (rejection of the null hypothesis) when *p* ≤ 0.05.

## 3. Results

### 3.1. Synthesis of Compounds 1-(2-Phenyltricyclo[3.3.1.1^3,7^]dec-2-yl)piperazine Maleate ***6*** and 1-[2-(4-Fluorophenyl)tricyclo[3.3.1.1^3,7^]dec-2-yl]piperazine Maleate ***7***

Compounds **6** and **7** were synthesized as outlined in Figure 1 and previously described by Fytas et al., 2015 [18]. Briefly, the 2-aryl-2-adamantyl chlorides (compounds **1** and **2**) were used as starting materials [21]. Each of these chlorides was heated with excess 1-Boc-piperazine to afford the 1-(*tert*-butoxycarbonyl)piperazine derivatives (compounds **3** and **4**) (S_N_1 reaction), respectively. The Boc-group was then removed with trimethylsilylchloride and sodium iodide to give the corresponding piperazines **6** and **7**. Alternatively, compound **7** was obtained by heating chloride 2 with excess 1-Cbz-Piperazine to give the Cbz-Piperazine derivative **5**, which was then subjected to Cbz-deprotection by hydrogenolysis [18].

### 3.2. 1-(2-Aryl-2-adamantyl)piperazine Derivatives Inhibit Cell Growth of Melanoma Cancer Cell Lines

The most sensitive cell lines to both compounds were found to be the two melanoma cell lines SK-MEL-28 and MDA-MB-435, followed by the triple-negative breast cancer cell line MDA-MB-231 and the pancreatic ductal adenocarcinoma cell lines AsPCa and BxPC-3 (Figure 2 and Figure 3). Compound **7** showed stronger cytostatic and anti-proliferative activity, as indicated by the low (≤10 μM) TGI and GI_50_ values across most of the cell lines tested (Table 2 and Table 3).

### 3.3. Compound ***7*** Induces Stronger and Sustained Cytotoxic Effects in Melanoma Cells Compared to Compound ***6***

Since melanoma cells showed high sensitivity to both compounds, we further investigated the viability effect of compounds **6** and **7** in MDA-MB-435 cells, which were found to be somewhat more sensitive to the activity of both compounds (Table 2 and Table 3), while they also possess more favorable features for the development of human-to-mouse xenografts [22,23]. As shown in Figure 4a, compound **6** significantly reduced cell viability at 20 μΜ after 24 h of treatment, while at 10 μΜ, despite the slightly reduced cell viability initially, no difference was detected at the end of the treatment period (48 h), suggesting a concentration-dependent response. In contrast, compound **7** displayed a more potent and sustained cytotoxic effect as viability was reduced by nearly 80% at 20 μM and remained significantly suppressed even at 10 μM after 48 h (Figure 4b). Of note, treatment with 10 μΜ of compound **7** reduced cell viability to the same extent as 20 μΜ but required a longer treatment period (48 h), indicating that the cytotoxic activity of this compound is both time and concentration dependent.

### 3.4. Compounds ***6*** and ***7*** Inhibit the Clonogenicity of MDA-MB-435 Cells

Compounds **6** and **7** were tested for their ability to inhibit the clonogenic potential of MDA-MB-435 cells (Figure 5). Compound **6** inhibited colony formation only at the 5 μΜ concentration, while 1 and 2.5 μΜ showed no effect compared to the control group. In contrast, compound **7** showed strong anti-clonogenic capacity at all tested concentrations. At 2.5 and 5 μΜ, compound **7** almost completely inhibited the clonogenic capacity of MDA-MB-435, while at 1 μΜ the number of colonies was higher compared to 2.5 and 5 μΜ but significantly lower compared to the control group. These findings suggest that both compounds inhibit colony formation in MDA-MB-435 cells at concentrations in the low micromolar range, with compound **7** showing higher potency compared to compound **6**.

### 3.5. Compounds ***6*** and ***7*** Induce Cell Death in Melanoma Cells via Apoptosis and Autophagy-Related Mechanisms

Since both **6** and **7** were found to induce cell death in melanoma cancer cells under the experimental conditions tested, we aimed to further investigate the mechanism of cell death. To this aim, we studied key protein markers associated with apoptosis and autophagy pathways, and the results are shown in Figure 6 and Figure 7.

Analysis of PARP cleavage, a marker of apoptosis, revealed that treatment with 10 μM of compounds did not change the protein levels of full-length and cleaved PARP, compared to the control (Figure 6a–d). However, treatment with 20 µΜ of both compounds resulted in a significant increase in cleaved PARP protein levels, suggesting activation of the apoptotic pathway at longer exposure and higher concentrations of the compounds.

To further confirm the induction of apoptosis, we also evaluated the levels of procaspase 3. As can be seen in Figure 6a–f, treatment of cells with 20 μM of the compounds for 24 h resulted in a substantial decrease in procaspase 3 levels, indicating the activation of caspase 3. Interestingly, this reduction in procaspase-3 levels appears to be more pronounced in the case of compound **7**. These results are consistent with the results for PARP levels.

To investigate whether autophagy contributes to compound-induced cell death, we analyzed LC3 A/B I and II protein levels. In MDA-MB-435 cells treated with compound **6**, both LC3 A/B I and II levels were increased as early as 12 h post-treatment, in both tested concentrations (Figure 7a,c). Similarly, treatment with 10 μΜ of compound **7** significantly increased the levels of LC3 A/B I and II compared to control (Figure 7b,d). In cells treated with 20 μΜ of compound **7,** LC3 A/B I levels were initially increased at 6 and 12 h post-treatment, but increased at 24 h of treatment, compared to the control. LC3 A/B II levels remained consistently elevated at all time-points following treatment with 20 μΜ of compound **7**. These dynamic changes in the protein levels of LC3 A/B I and II suggest a potential involvement of autophagy in the observed cell death.

In light of the observed changes in LC3 II levels, we proceeded to evaluate the expression of GβL, a regulator of the mTOR pathway, in order to investigate the potential interaction of these compounds with the mTOR/PI3K/Akt axis at the structural/regulatory level. Under the tested experimental conditions, compound 6 did not alter GβL protein levels compared to the control (see Figure 7a,e), whereas treatment with 20 μM of compound **7** reduced GβL protein levels, though this reduction did not reach statistical significance (see Figure 7b,f). These findings suggest that the compounds may not directly modulate the mTOR kinase complex. However, the precise role of the mTOR/PI3K/Akt pathway in the activity of these (adamantyl)piperazines remains unclear.

## 4. Discussion

The aim of this study was to characterize the cytotoxic and anti-proliferative properties of two 1-(2-aryl-2-adamantyl)piperazine derivatives, **6** and **7**, with particular emphasis on their effects on melanoma cells. The significance of this study is further underscored by the global rise in cancer incidence, specifically melanoma, one of the most diagnosed cancers in the EU in both sexes [2]. Melanoma was ranked fourth among cancers in women and sixth among cancers in men in terms of incidence and mortality in the EU in 2022 [2].

In a previous study [18], several novel 1-(2-aryl-2-adamantyl)piperazines were synthesized and evaluated for their in vitro anti-proliferative activities. Among these, two compounds, piperazine **6** and its fluorinated derivative **7** showed the most promise in vitro anti-proliferative activities.

In the present study, we further evaluated the in vitro anticancer activity of these two promising new piperazines across a panel of human cancer cell lines. Our results show that both compounds reduced cell viability and proliferation against various human cancer cell lines derived from various types of cancers, with melanoma cells showing the highest sensitivity. Subsequent experiments demonstrate concentration-dependent effects on the viability and proliferation of MDA-MB-435 melanoma cells, establishing compound **7** as more potent than **6**. Clonogenicity/long-term experiments further support these findings, showing that both compounds effectively inhibit colony formation at low μΜ concentrations, suggesting compound **7** to be more efficient than **6**. Of note, although the effects were not directly evaluated in normal cells or in vivo, the two agents exhibited differential activity rather than uniform, high toxicity across all cell lines tested (see Table 2 and Table 3 for LC_50_ values and the GI_50_ of compound **6**). Since the current study is largely based on cell lines included in the NCI-60 cell line panel, it is feasible to get some idea of the general toxicity of the compounds, suggesting these two agents most probably do not have a generally toxic profile [24].

The two compounds **6** and **7** have similar pKa values, with compound **7** being slightly more lipophilic (ClogP = 4.58 vs. 4.43, Table 4).

This modest increase, however, can enhance passive membrane permeability and facilitate greater intracellular accumulation, especially in the acidic tumor microenvironment, where weak bases undergo ion trapping. As a result, compound **7** achieves higher intracellular exposure, which plausibly explains its superior anticancer activity. Another significant difference between the two molecules is the presence of a fluoride atom in the phenyl group, which could explain the superior activity of compound **7**. Fluorination is a strategy used in designing effective anticancer drugs, as fluorine atoms may enhance drug stability, lipophilicity, and cell permeability, and facilitate interactions with biological targets [25], even enhancing sigma receptor affinity [26]. Fluorinated compounds like 5-fluorouracil and approved drugs such as gefitinib (quinazoline, which is substituted by a (3-chloro-4-fluorophenyl) nitrilo group, a 3-(morpholin-4-yl)propoxy group, and a methoxy group at positions 4, 6, and 7, respectively) and capecitabine (cytidine in which the hydrogen at position 5 is replaced by fluorine and the amino group attached to position 4 is converted into its N-(penyloxy) carbonyl derivative) are used to treat various cancers. Nonetheless, direct mechanistic links need further detailed investigation.

To explore the underlying mechanisms of cell death, we analyzed markers of apoptosis and autophagy in MDA-MB-435 cells treated with the compounds. Our current data demonstrate an increase in LC3 II levels, indicating the induction of autophagy. However, we acknowledge that this alone is insufficient to discriminate between increased autophagosome formation and impaired autophagic degradation. Moreover, the autophagic pathway may play a dual role: cell rescue initially, which is more evident at a concentration of 10 μM and with derivative **6**, where cells can recover, as shown by the trypan blue results (Figure 3a); and cell death at higher concentrations later on, probably due to irreversible damage caused by high concentrations of compound **6** and both concentrations of compound **7**. Apparently, further studies addressing autophagic flux (autophagic degradation) with the use of inhibitors like bafilomycin A1 or chloroquine (which block lysosomal degradation) are required to confirm the full induction and the role of autophagy in the cell death mechanism of compounds **6** and **7**.

Given the central role of the mTOR/PI3K/Akt signaling axis in regulating autophagy and cancer in general [27], we examined the effects of these two compounds on the expression of the mTOR pathway regulator GβL/mLST8. This analysis aimed to primarily explore potential interactions of our compounds with the mTOR complex at a structural/regulatory level and provide some preliminary insight into whether modulation of this pathway contributes to the anticancer effects observed with the compounds. GβL/mLST8 protein (G protein β subunit-like/mammalian lethal with SEC13 protein 8) is a core regulatory subunit of both mTOR complexes, mTORC1 and mTORC2, that plays a pivotal role in mTOR kinase activity. In cultured human cancer cells, depletion of GβL/mLST8 has been shown to impair mTORC1 signaling [28,29], suggesting that GβL/mLST8 may have a role in cancer via modulating mTORC1/C2 complexes’ function. Although compounds **6** and **7** were not found to significantly alter GβL/mLST8 protein levels under the experimental conditions of this study, targeting GβL/mLST8 could be a promising approach to improving the effectiveness of mTOR inhibitors. In light of the data showing that modulation of impaired mTORC1 signaling can result in destabilized mTOR complexes, modulation of GβL/mLST8 might lead to impaired mTOR signaling, making cancer cells more susceptible to mTOR-targeted therapies by promoting cell death pathways such as apoptosis or autophagy. Clearly, further studies investigating the combined effects of (adamantyl)piperazines and mTOR inhibitors are needed to determine whether GβL contributes to therapeutic responsiveness. Additionally, we must acknowledge that GβL is an indirect marker of mTOR activity. Future studies using downstream effectors, such as phospho-S6K or 4E-BP1, are necessary in order to assess mTOR kinase activity directly in response to these compounds.

In a subsequent study, combined evidence from caspase 3 and PARP cleavage supports the induction of apoptosis by both agents. The activation of this pathway was again found to be concentration and time dependent. The increased levels of LC3 II and the activation of apoptosis may indicate a pleiotropic mechanism of cell death for these compounds, which requires further clarification, especially regarding the role of autophagy, as previously mentioned. However, as with autophagy, future work needs to extend these findings by including upstream caspase assays to substantiate this pathway further and reveal the exact apoptotic pathway activated.

Interestingly, adamantane analogues have been reported as anti-proliferative and cytotoxic agents, while the cytotoxic activity was found to be related to their affinity for sigma receptors [8,15,16,17]. Sigma1 and sigma2 receptors have drawn attention to their potential therapeutic implications despite an incomplete understanding of their mechanisms of action. Sigma2 receptors are being explored as targets for cancer therapy and imaging agents and have been associated with the induction of autophagy [30] and apoptosis in cancer cells [31,32]. Sigma1 has also been implicated in promoting cellular survival under oxidative stress by transcriptionally regulating Bcl-2 via the ROS-NF-κB pathway [33]. In addition, several studies have shown that sigma2 ligands can induce cell death in several cancer types through caspase-dependent and -independent apoptosis, lysosomal leakage, oxidative stress, Ca^2+^ release, ceramide production, autophagy, and cell cycle disruption [34,35,36,37,38,39,40,41]. Given the affinity of our compounds to sigma receptors, and the reported overexpression of these receptors in MDA-MB-435 cells [20], we hypothesize that the observed effects of these (adamantyl)piperazines may be mediated, at least in part, through sigma receptors. This hypothesis remains to be validated, and ongoing mechanistic studies of our group are directed toward elucidating these molecular interactions by using genetically modified cancer cells.

A major limitation we must acknowledge in this study is the use of MDA-MB-435 as our sole model for studying the mechanisms of action. Firstly, its origin is debated [42,43]. Secondly, it is not fully representative of melanoma. MDA-MB-435 is reported to be representative of amelanotic melanoma, which limits our ability to study the role of melanogenesis in melanoma progression and management, given that melanogenesis is reported to be closely linked to the advancement of melanotic melanoma [44].

Nevertheless, MDA-MB-435 cells remain an accepted model in several areas of tumor biology. For example, they are still used as a melanoma model in the NCI-60 panel and are employed by the COMPARE algorithm to draw significant conclusions [19]. Furthermore, a wealth of information is available from the extensive analysis of the cell lines included in the NCI-60 panel, while it has more favorable features over SK-MEL-28 for developing in vivo models [22,23], which are necessary for further evaluation of the activity of these (adamantyl)piperazine derivatives.

## 5. Conclusions

In conclusion, we present two novel 1-(2-aryl-2-adamantyl)piperazine derivatives, **6** and **7**, which exhibit significant cytotoxic and anti-proliferative effects, particularly against melanoma cells. Among them, the fluorinated derivative **7** showed strong potency, highlighting the impact of the fluorination at the phenyl ring on biological activity. This study suggests that these compounds hold promise as potential therapeutic agents, especially against melanoma. However, since these findings have only been observed in MDA-MB-435 melanoma cells, future studies must validate them in additional melanoma cell lines with different genetic and phenotypic characteristics to elucidate the precise molecular mechanisms involved.

## Figures and Tables

**Figure 1 medicina-61-01731-f001:**
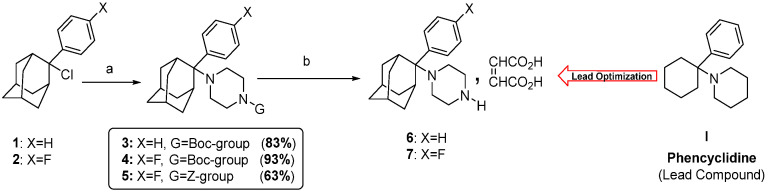
Synthetic approach for the development of compounds **6** and **7**: (a) Appropriate 1-substituted piperazine (excess), 105 °C, 7 h for **3** and **4** or 105 °C, 10 h for **5**, argon; (b) Me_3_SiCl, NaI, CH_3_CN/CHCl_3_ 2/1, RT, 1 h, argon, for **6** from **3** and for **7** from **4** or H_2_/Pd-C, EtOH-AcOEt 1:3 (*v*/*v*), RT, 3 h, for **7** from **5**.

**Figure 2 medicina-61-01731-f002:**
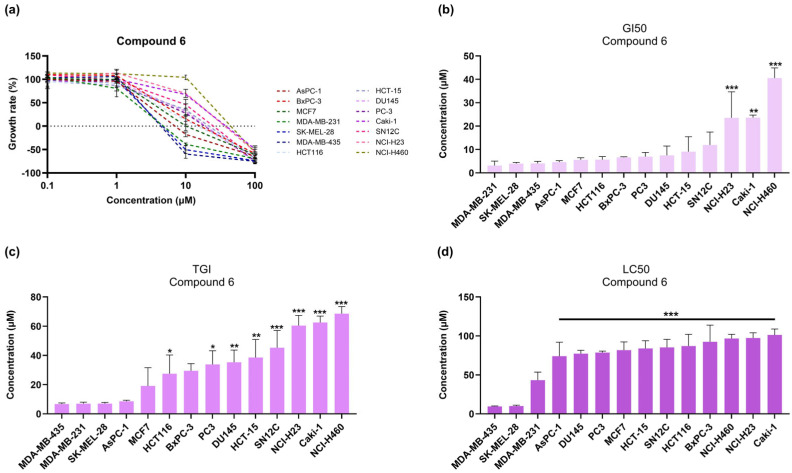
Dose–response effect of compound **6** on cell growth. (**a**) Graph displaying the effect of 0.1, 1, 10, and 100 μΜ of compound **6** on the growth rate of established cell lines of the NCI-60 human tumor cell line panel. (**b**–**d**) Bar graphs showing the GI_50_, TGI, and LC_50_ parameters calculated for every cell line treated with compound **6**, in comparison to MDA-MB-435 values. The statistical test performed was one-way ANOVA, with Holm–Šidák post hoc test. Each point or bar represents the average of n ≥ 2 independent experiments ± SEM. Statistical significance is indicated as follows: *p* = 0.033 (*), *p* = 0.002 (**), and *p* < 0.001 (***) vs. MDA-MB-435.

**Figure 3 medicina-61-01731-f003:**
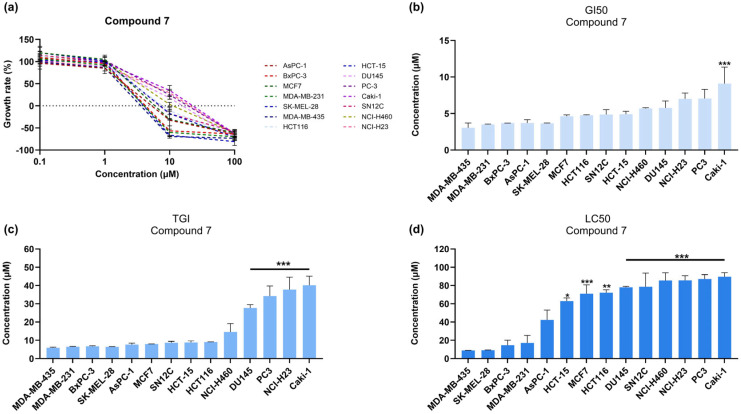
Dose–response effect of compound **7** on cell growth. (**a**) Graph displaying the effect of 0.1, 1, 10, and 100 μΜ of compound **7** on the growth rate of established cell lines of the NCI-60 human tumor cell line panel. (**b**–**d**) Bar graphs showing the GI_50_, TGI, and LC_50_ parameters calculated for every cell line treated with compound **7**, with statistical significance in comparison to MDA-MB-435 values. The test performed was one-way ANOVA, with a Holm–Šidák post hoc test. Each point or bar represents the average of n ≥ 2 independent experiments ± SEM. Statistical significance is indicated as follows: *p* = 0.033 (*), *p* = 0.002 (**), and *p* < 0.001 (***) vs. MDA-MB-435.

**Figure 4 medicina-61-01731-f004:**
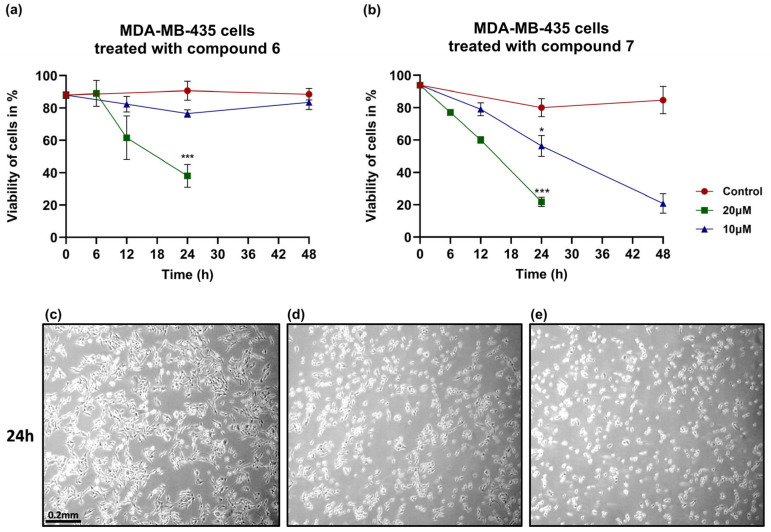
Cell viability of MDA-MB-435 is impaired when incubated with compounds **6** and **7**. (**a**,**b**) Graphs illustrating the effect of 10 and 20 μΜ of compound **6** and **7**, respectively, on MDA-MB-435 cell viability. (**c**–**e**) From left to right, the pictures depict control MDA-MB-435 cells, MDA-MB435 cells treated with 10 μΜ of compound **6**, and cells treated with 10 μΜ of compound **7**, magnification 5×Statistical analysis was performed in comparison to the MDA-MB-435 cells. The test performed was one-way ANOVA, with a Holm–Šidák post hoc test. Each point in (**a**,**b**) represents the average of *n* ≥ 3 independent experiments ± SEM. Statistical significance is indicated as follows: *p* = 0.033 (*), and *p* < 0.001 (***) vs. control.

**Figure 5 medicina-61-01731-f005:**
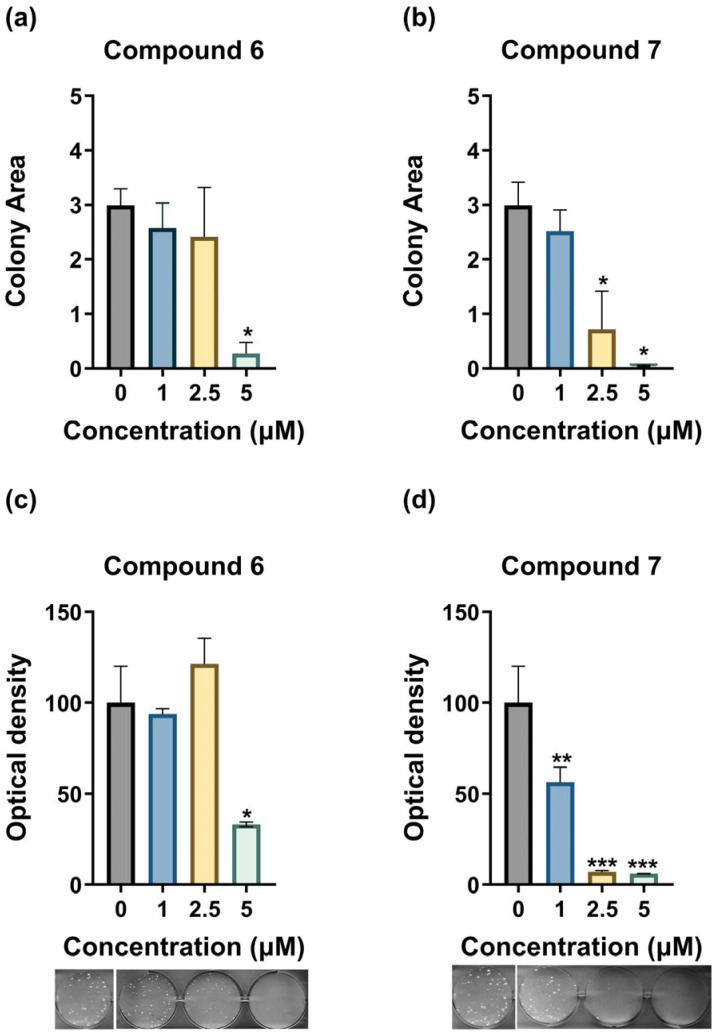
Treatment of MDA-MB-435 cells with compounds **6** and **7** affects colony formation. (**a**,**b**) Bar graphs showing the colonies’ area size when MDA-MB-435 cells were treated with compounds **6** and **7**, compared to untreated (control) cells. To measure the area size, we utilized ImageJ (Fiji version 1.54p). (**c**,**d**) The bar graphs illustrate the optical density of treated and untreated MDA-MB-435 cells after their fixation and staining with SRB, with representative pictures of each condition under the graphs. For the statistical analysis, we performed one-way ANOVA, with a Holm–Šidák post hoc test. Each bar represents the average of n ≥ 2 independent experiments ± SEM. Statistical significance is indicated as follows: *p* = 0.033 (*), *p* = 0.002 (**), and *p* < 0.001 (***) vs. control.

**Figure 6 medicina-61-01731-f006:**
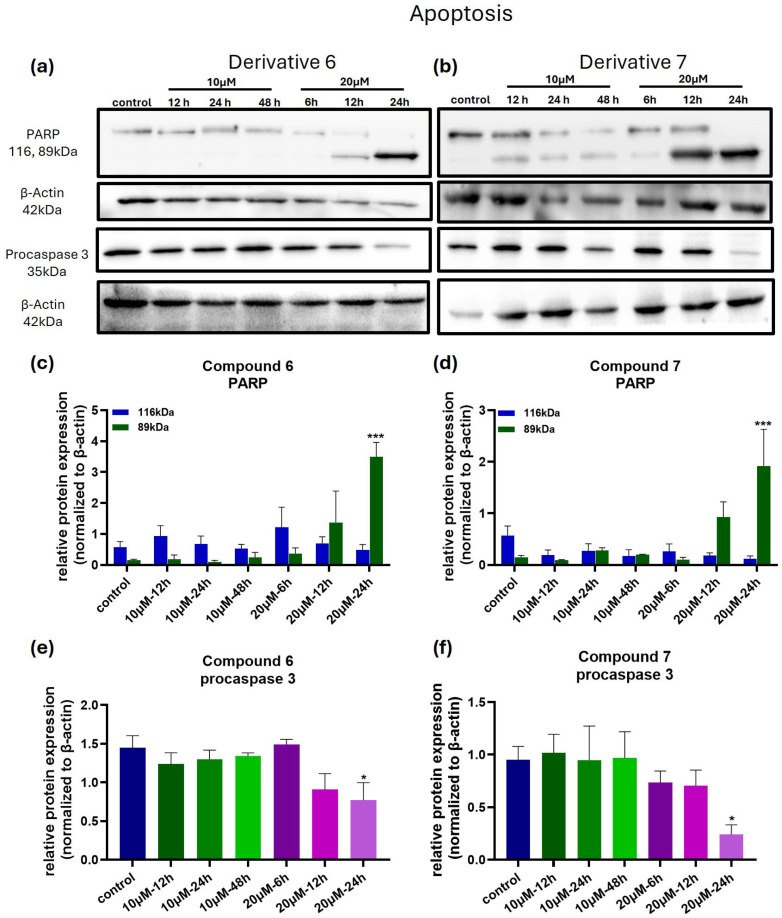
Compounds **6** and **7** induce cellular apoptosis. (**a**,**b**) Western blot membranes depicting the reduction in protein levels of full-length PARP (116 KDa), the increased protein levels of cleaved PARP (89 KDa), and procaspase 3 protein levels with lysates of MDA-MB-435 cells isolated from the trypan blue viability assay. (**c**,**d**) The bar graphs illustrate the densitometry of the Western blot experiments regarding PARP expression in MDA-MB-435 cells treated with compounds **6** and **7**, respectively. (**e**,**f**) Bar graphs showing procaspase 3 protein levels in MDA-MB-435 cells treated with compounds **6** and **7**. For the statistical analysis, we performed one-way ANOVA, with a Holm–Šidák post hoc test. Each bar represents the average of n ≥ 2 independent experiments ± SEM. Statistical significance is indicated as follows: *p* = 0.033 (*), and *p* < 0.001 (***) vs. control cells.

**Figure 7 medicina-61-01731-f007:**
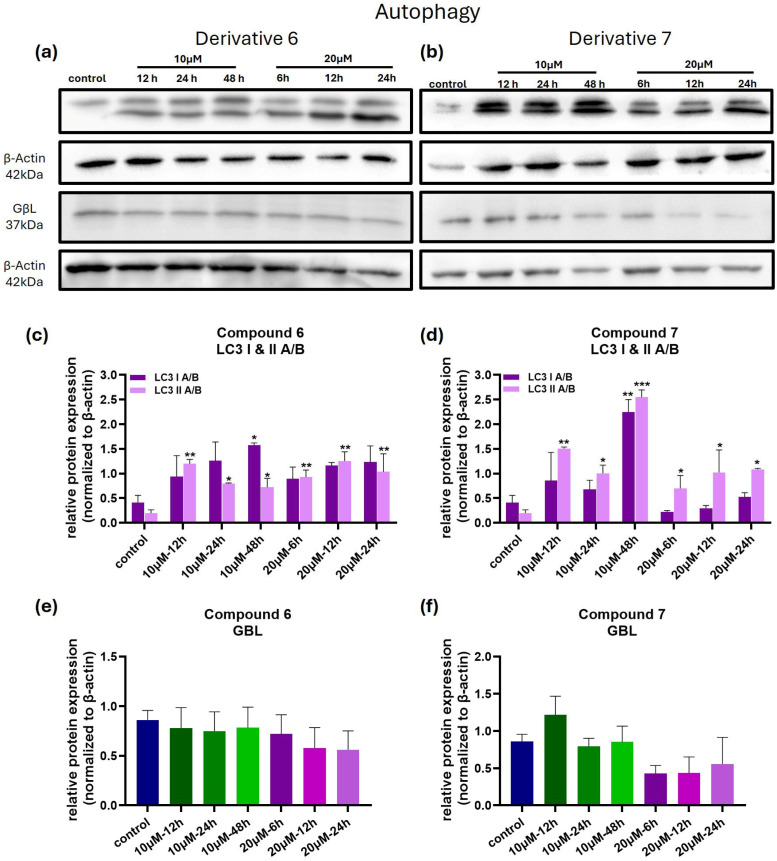
Compounds **6** and **7** regulate autophagy in MDA-MB-435 cells. (**a**,**b**) Western blot membranes illustrate the activation of the autophagy pathway in MDA-MB-435 cells treated with compounds **6** and **7**. (**c**,**d**). Bar graphs illustrating the LC3 A/B I and II protein levels when MDA-MB-435 cells were treated with compounds **6** and **7**. (**e**,**f**) Bar graphs showing GBL protein levels in MDA-MB-435 cells treated with compounds **6** and **7**. The test performed was one-way ANOVA, with a Holm–Šidák post hoc test. Each bar represents the average of n ≥ 2 independent experiments ± SEM. Statistical significance is indicated as follows: *p* = 0.033 (*), *p* = 0.002 (**), and *p* < 0.001 (***) vs. control cells.

**Table 1 medicina-61-01731-t001:** Human cancer cell lines used in this study, classified according to cancer type.

Cancer Type	Cell Lines
Pancreas	AsPC-1, BxPC-3
Breast	MCF7, MDA-MB-231
Melanoma	SK-MEL-28, MDA-MB-435
Colorectal	HCT116, HCT15
Prostate	DU145, PC3
Renal	Caki-1, SN12C
Lung	NCI-H23, NCI-H460

**Table 2 medicina-61-01731-t002:** The concentrations of compound **6**, which are required to achieve growth inhibition in 50% of the cell population (GI_50_), total growth inhibition (TGI), and cell death in 50% of the cell population (LC_50_), for all the cell lines tested. Melanoma cell lines are indicated in bold.

	GI50 (μΜ)	TGI (μΜ)	LC50 (μΜ)
AsPC-1	4.6	8.6	72.1
BxPC-3	6.6	28.7	89.6
MCF7	5.6	19.1	83.8
MDA-MB-231	3.3	6.9	43.2
**SK-MEL-28**	**4.0**	**7.0**	**10.1**
**MDA-MB-435**	**4.0**	**6.8**	**9.5**
HCT116	5.8	25.4	86.2
HCT-15	7.4	38.6	83.3
DU145	7.4	36.2	77.5
PC-3	6.9	34.6	78.8
Caki-1	23.6	62.3	100.9
SN12C	11.9	45.9	85.0
NCI-H23	24.6	60.8	97.0
NCI-H460	40.5	68.5	96.6

**Table 3 medicina-61-01731-t003:** The concentrations of compound **7**, which are required to achieve growth inhibition in 50% of the cell population (GI_50_), total growth inhibition (TGI), and cell death in 50% of the cell population (LC_50_), for all the cell lines tested. Melanoma cell lines are indicated in bold.

	GI50 (μΜ)	TGI (μΜ)	LC50 (μΜ)
AsPC-1	3.7	7.5	42.4
BxPC-3	3.7	6.7	14.6
MCF7	4.6	7.9	67.4
MDA-MB-231	3.5	6.4	17.2
**SK-MEL-28**	**3.9**	**6.9**	**9.8**
**MDA-MB-435**	**3.1**	**6.0**	**8.9**
HCT116	4.7	9.0	71.6
HCT-15	4.9	8.7	64.2
DU145	6.0	27.9	78.0
PC-3	7.1	35.3	86.3
Caki-1	9.1	42.0	88.1
SN12C	4.9	8.6	78.6
NCI-H23	6.9	38.1	85.9
NCI-H460	5.7	14.5	83.9
Caki-1	3.7	7.5	42.4
SN12C	3.7	6.7	14.6
NCI-H23	4.6	7.9	67.4
NCI-H460	3.5	6.4	17.2

**Table 4 medicina-61-01731-t004:** In silico-predicted physicochemical parameters (logP and pKa) for compounds **6** and **7**.

	CLogP	pKa
Compound **6**	4.434	8.956
Compound **7**	4.577	8.950

## Data Availability

Data are available upon reasonable request.

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
