# Peer review of "Novel 1-(2-Aryl-2-adamantyl)piperazine Derivatives Exhibit In Vitro Anticancer Activity Across Various Human Cancer Cell Lines, with Selective Efficacy Against Melanoma"

_medicina, 2025, doi:10.3390/medicina61101731_

Round 1

Reviewer 1 Report

Comments and Suggestions for Authors

This manuscript by Papapostolou et al described the profiling of the growth inhibition effects of two previously reported synthetic compounds over a panel of cancer cell lines. The author found that compound 7 showed most potent effect on melanoma cell line MDA-MB-435. Western blot results indicate the cell death involves both apoptotic and autophagic pathways (questionable).

The following points must be addressed:

  1. The conclusion about the involvement of autophagic death is not supported by current data. The western blot of LC3-II experiment should include a lysosomal inhibitor (Bafilomycin A1) in order to draw any meaningful conclusion.
  2. The title of the manuscript must be changed (to remove autophagy part), because of the reason stated in point #1.
  3. The whole introduction section needs to be re-written. There are a lot of words about sigma receptors, which are completely irrelevant to the experimental data presented in this work (the inclusion of sigma receptors in the discussion section is fine). Also, line46-48 about computation/AI is also completely irrelevant.
  4. An important limitation of this work is that it did not include any general toxicity studies, whether it’s with normal non-cancer cells or in vivo. I understand that this may be beyond the scope of this manuscript, however, relevant discussions should be included.
  5. Numbers of replicates should be noted in each figure. Also, in figure 5a, why there’s so much variation in data compared to figure 5b?
  6. Line203-204 and figure 1: “Z group” is not a standard abbreviation. Did the author mean “Cbz” ?
  7. The Materials and Methods section needs to be improved:
  8. Line118, “predetermined densities”. What are the numbers?
  9. Line121, “subset of cells”, how much fraction was that?
  10. Line155, “13.000 rpm”, should be 13000 I assume?
  11. Line181, “medium renewed every 7 days”. The renewed medium also contained corresponding doses of testing compounds, is that correct?

Author Response

Dear Editor and Reviewers,

We would like to express our sincere gratitude for the time and effort you have devoted to reviewing our manuscript, which has been of great assistance in strengthening our work.

In response to concerns raised by the editor and reviewers, the manuscript has been heavily revised to address all comments and correct other minor errors. All changes have been tracked and highlighted.

We hope that we have properly addressed all of your concerns.

Please see our point-by-point responses to your comments below.

This manuscript by Papapostolou et al described the profiling of the growth inhibition effects of two previously reported synthetic compounds over a panel of cancer cell lines. The author found that compound 7 showed most potent effect on melanoma cell line MDA-MB-435. Western blot results indicate the cell death involves both apoptotic and autophagic pathways (questionable).

The following points must be addressed:

  1. The conclusion about the involvement of autophagic death is not supported by current data. The western blot of LC3-II experiment should include a lysosomal inhibitor (Bafilomycin A1) in order to draw any meaningful conclusion.

Authors' reply: We thank the reviewer for this important observation regarding our interpretation of the autophagy data. We agree that the involvement of autophagic cell death cannot be conclusively established without experiments that assess autophagic flux, such as the inclusion of lysosomal inhibitors (e.g., Bafilomycin A1) in the LC3-II western blot assay. Our current data demonstrate an increase in LC3-II levels, which indicates induction of autophagy, but we acknowledge that this alone is insufficient to discriminate between increased autophagosome formation and impaired autophagic degradation. Accordingly, in the revised manuscript we have tempered our conclusion to state that our findings suggest an autophagic response, but additional experiments are required to confirm whether this process contributes directly to cell death. We also highlight this limitation in the discussion and propose follow-up studies using Bafilomycin A1 or other inhibitors to rigorously assess autophagic flux.

Results, 325-326: These dynamic changes in the protein levels of LC3 A/B I and II suggest a potential involvement of autophagy in the observed cell death.

Discussion, lines 405-417: To explore the underlying mechanisms of cell death, we analyzed markers of apoptosis and autophagy in MDA-MB-435 cells treated with the compounds. Our current data demonstrate an increase in LC3 II levels, indicating the induction of autophagy. However, we acknowledge that this alone is insufficient to discriminate between increased autophagosome formation and impaired autophagic degradation. Moreover, the autophagic pathway may play a dual role: cell rescue initially, which is more evident at a concentration of 10 μM and with derivative 6, where cells can recover, as shown by the trypan blue results (Fig. 3a); and cell death at higher concentrations later on, probably due to irreversible damage caused by high concentrations of compound 6 and both concentrations of compound 7. Apparently, further studies addressing autophagic flux (autophagic degradation) with the use of inhibitors like bafilomycin A1 or chloroquine (which block lysosomal degradation) are required to confirm the full induction and the role of autophagy in the cell death mechanism of compounds 6 and 7.

Discussion, lines 442-446: The activation of this pathway was again found to be concentration- and time-dependent. The increased levels of LC3 II and the activation of apoptosis may indicate a pleiotropic mechanism of cell death for these compounds, which requires further clarification, especially regarding the role of autophagy, as previously mentioned.

  1. The title of the manuscript must be changed (to remove autophagy part), because of the reason stated in point #1.

Authors' reply: The title of the manuscript has been changed to eliminate any potential misinterpretation by readers, and to better represent the work as a whole:

Current title: Novel 1-(2-aryl-2-adamantyl)piperazine derivatives exhibit in vitro anticancer activity across various human cancer cell lines, with selective efficacy against melanoma.

3. The whole introduction section needs to be re-written. There are a lot of words about sigma receptors, which are completely irrelevant to the experimental data presented in this work (the inclusion of sigma receptors in the discussion section is fine). Also, line46-48 about computation/AI is also completely irrelevant.

Authors' reply: We thank the reviewer for this constructive feedback on the introduction. We agree that the current version devotes disproportionate attention to sigma receptors, which are not directly linked to the experimental data presented. In response to this comment as well as the relevant comments raised by the Editor, we have substantially revised whole introduction to focus more closely on the biological problem under investigation and the rationale for our experimental approach. The discussion of sigma receptors has been reduced and relocated to the discussion section, where it serves as broader context for potential implications of our findings. Additionally, the sentences about computation/AI (lines 46–48) have been removed. We believe these changes improve the focus and clarity of the introduction and ensure closer alignment with the experimental work presented.

4. An important limitation of this work is that it did not include any general toxicity studies, whether it’s with normal non-cancer cells or in vivo. I understand that this may be beyond the scope of this manuscript, however, relevant discussions should be included.

Authors' reply: We thank the reviewer for raising this important point. We believe as the reviewer states that it is quite premature to talk about toxicity at this point in the study.  We aim to test toxicity when we proceed with the in vivo evaluation of the compounds’ efficacy. However, since the current study is largely based on cell lines included in the NCI-60 cell line panel it is feasible to get some idea of the general toxicity of the compounds based on previously published works [24].  Thus, we have added a part in the discussion section on the potential general toxicity of the compounds under study.

Discussion lines 371-374: Of note, although the effects were not directly evaluated in normal cells or in vivo, the two agents exhibited differential activity rather than uniform, high toxicity across all cell lines tested (see Tables 2 and 3 for LCâ‚…â‚€ values and the GIâ‚…â‚€ of compound 6). Since the current study is largely based on cell lines included in the NCI-60 cell line panel, it is feasible to get some idea of the general toxicity of the compounds suggesting most probably not a generally toxic profile of these two agents [24].

  1. Numbers of replicates should be noted in each figure. Also, in figure 5a, why there’s so much variation in data compared to figure 5b?

Authors' reply: We thank the reviewer for this helpful comment. We have now added the number of replicates to each figure legend in the manuscript. We apologise for the discrepancy between Figures 5a and 5b. The analysis was inappropriate, as the controls (untreated cells) should obviously be the same in both cases.  We have now corrected this discrepancy and included an updated version of Figure 5, which better highlights our findings. Panels a and b show the colony area measured using ImageJ after incubating MDA-MB-435 cells with compounds 6 and 7. Panels c and d present the optical density of the cells following fixation and staining with SRB.

6. Line203-204 and figure 1: “Z group” is not a standard abbreviation. Did the author mean “Cbz”?

Authors' reply: We thank the reviewer for this comment. The reviewer is right. In the revised manuscript, we have clarified that the abbreviation “Z” refers to the benzyloxycarbonyl protecting group, which is commonly denoted as “Cbz” in the literature. To avoid confusion, we have replaced “Z” with “Cbz” throughout the text and in Figure 1, in line with the IUPAC-recommended and widely accepted nomenclature.

Results, lines 217-220: Alternatively, compound 7 was obtained by heating chloride 2 with excess 1-Cbz-Piperazine to give the Cbz-Piperazine derivative 5, which was then subjected to Cbz-deprotection by hydrogenolysis [18].

 The Materials and Methods section needs to be improved:

7. Line 118, “predetermined densities”. What are the numbers?

Authors' reply: We have added a reference that is related to the inoculation densities of the cells.

Materials and methods, lines 124-126: For AcPC-1 and BxPC-3 the plating densities were 5.000 each, while for the rest, as it has been reported in previous works [19].

  1. Line 121, “subset of cells”, how much fraction was that?

Authors' reply: We have clarified this point in the revised manuscript.

Results, lines 151-154: Before adding the compounds, 100 μl of the culture from each flask was collected at time zero (T0). The samples were mixed 1:1 with a 0.5% (w/v) trypan blue solution (Biochrom, L6323), incubated at room temperature for 2–3 minutes, and then assessed for cell viability using a haemocytometer under a microscope.

9. Line 155, “13.000 rpm”, should be 13000 I assume?

Authors' reply: This is correct, thank you for pointing out this error. It has been corrected in the revised version of our manuscript

Materials and methods, lines 168-169: …minutes and centrifuged at 1300 rpm for 30 minutes at 4oC.

10. Line181, “medium renewed every 7 days”. The renewed medium also contained corresponding doses of testing compounds, is that correct?

Authors’ reply: This is correct, we have clarified this in the revised version of the manuscript.

Materials and Methods, lines 194-196: The cells were cultured for 15 days, with the medium (supplemented with the compounds, where applicable) refreshed every seven days.

Reviewer 2 Report

Comments and Suggestions for Authors
  1. The title claims induction of bothautophagic and apoptotic cell death, but apoptosis data (e.g., PARP cleavage) is weaker than autophagy (LC3-II). Either strengthen apoptosis evidence or refine the title.
  2. The abstract overemphasizes autophagy but underrepresents apoptosis data. Include specific apoptosis markers (e.g., caspase-3 cleavage) for balance.
  3. The link between sigma receptors and adamantane derivatives is speculative. Provide direct references to prior evidence (or state this is a novel hypothesis).
  4. Why prioritize MDA-MB-435 (historically controversial for melanoma origin) over SK-MEL-28? Justify or address potential misclassification.
  5. SRB measures total protein, not viability. Clarify how it distinguishes cytostasis vs. cytotoxicity (e.g., via LC50 vs. GI50).
  6. LC3-II accumulation alone doesn’t prove autophagy; include flux assays (e.g., bafilomycin A1) to confirm autophagosome turnover.
  7. Caspase-3 cleavage is absent despite PARP data. Include caspase-3/9 assays to solidify apoptosis claims.
  8. GBL is an atypical marker for mTOR. Use phospho-S6K/S6 or 4E-BP1 to directly assess mTOR activity.
  1. Dose-Response Curves-Fig. 2/3 lack error bars or statistical significance markers. Add SEM and p-values for key comparisons.
  2. Clonogenic Assay-Colony counts are qualitative (images) but quantified via SRB absorbance. Use direct colony counting for rigor.
  1. Western Blot Quantification-Densitometry graphs (Figs. 6–7) lack normalization controls (e.g., actin for PARP, GBL). Reanalyze with proper loading controls.
  1. Time-Course Gaps-Autophagy/apoptosis markers are shown at limited timepoints (e.g., 6–24h). Extend to 48h to align with viability assays.
  2. The claim that fluorination enhances potency (p. 12) lacks mechanistic support. Compare logP/pKa or sigma receptor binding to justify.
  3. No experimental evidence ties effects to sigma receptors. At least cite overexpression in MDA-MB-435 (Ref. 11) or propose knockdown experiments.
  4. The interplay is oversimplified. Discuss if autophagy is pro-survival (early) or pro-death (late), citing flux vs. apoptosis timing.
  5. Figure 4: Label microscopy images with scale bars.
  6. Table 2/3: Highlight melanoma-specific results (bold/asterisks) for clarity.
  7. Typos: "pro-Caspase 3" → "procaspase-3" (consistent capitalization).

Author Response

Dear Editor and Reviewers,

We would like to express our sincere gratitude for the time and effort you have devoted to reviewing our manuscript, which has been of great assistance in strengthening our work.

In response to concerns raised by the editor and reviewers, the manuscript has been heavily revised to address all comments and correct other minor errors. All changes have been tracked and highlighted.

We hope that we have properly addressed all your concerns.

Please see our point-by-point responses to your comments below.

  1. The title claims induction of both autophagic and apoptotic cell death, but apoptosis data (e.g., PARP cleavage) is weaker than autophagy (LC3-II). Either strengthen apoptosis evidence or refine the title.

Authors’ reply: We thank the reviewer for this insightful comment. In the revised version of the manuscript, we have done both. We have revised the title of the manuscript to eliminate any potential misinterpretation by readers, and to better represent the work as a whole and strengthen the data on apoptosis by adding procaspase 3 expression levels (see results and figure…

Current title: Novel 1-(2-aryl-2-adamantyl)piperazine derivatives exhibit in vitro anticancer activity across various human cancer cell lines, with selective efficacy against melanoma.

  1. The abstract overemphasizes autophagy but underrepresents apoptosis data. Include specific apoptosis markers (e.g., caspase-3 cleavage) for balance.

Authors’ reply: We thank the reviewer for this important observation. We agree that the abstract placed more emphasis on autophagy, while the apoptosis data were underrepresented. In the revised manuscript, we have updated the abstract to include caspase-3 results as well while we provide a more balance overview of our observations to address at the same time the concerns raised for the autophagy results.

Abstract, lines 26-32: Furthermore, western blot experiments indicate that both 6 and 7 induce LC3 accumulation, procaspase 3 decrease and PARP cleavage, suggesting the implication of multiple death pathways in their anticancer mechanism of action. Conclusions: This study sheds light on the in vitro anticancer potential of two novel 1-(2-aryl-2-adamantyl) piperazine derivatives. It highlights their differential activity against melanoma and emphasizes their potential as lead candidates for further therapeutic exploration.

  1. The link between sigma receptors and adamantane derivatives is speculative. Provide direct references to prior evidence (or state this is a novel hypothesis).

Authors’ reply: We thank the reviewer for this comment. We agree that the connection between sigma receptors and adamantane derivatives in our study is largely exploratory. To address this, we have now cited previous studies demonstrating the interaction between adamantane-based compounds and sigma receptors (see relevant references). We have added this clarification to the Introduction and Discussion sections to make it clear to readers that this link is speculative.

Introduction, lines 67-74: Adamantane derivatives have been previously reported to show potent anticancer activities in vitro and in vivo against a variety of human cancer cell lines and against human to mouse xenografts [15–17]. C1-substituded adamantanes were reported to show in vitro activity against a panel of human cancer cell lines (including the melanoma cells LOX-IMVI) and in vivo activity against pancreatic cancer xenografts [15], while adamantane phenylalkylamines with σ-receptor binding affinity have been reported to show potent antimetastatic activity against the human PC3 prostate cancer xenograft together with analgesic activities [17].

  1. Why prioritize MDA-MB-435 (historically controversial for melanoma origin) over SK-MEL-28? Justify or address potential misclassification.

Authors’ reply: MDA-MB-435 cells were selected because they were found to be somewhat more sensitive to the activity of both compounds 6 and 7. On the other hand, MDA-MB-435 cells show more favorable characteristics for use as in vivo melanoma model as compared to the other melanoma cell line we used in this study the SK-MEL-28. Finally, this later cell line was tested in vivo in our lab and found totally inacceptable for in vivo experiments as the corresponding xenografts did not grow well (reaching a size of less than 400mm3 in a period of 3 months after the inoculation of the cells in mice) despite the fact we used NOD/SCID mice with a much severe immunocompromised phenotype as compared to other strains like for example the nude mice (this is in agreement with already published data, please see DOI: 10.1155/2023/5568206 and https://doi.org/10.1016/j.nima.2006.10.051).  These details have been also added to the results section.

Results lines 258-262: Since melanoma cells showed high sensitivity to both compounds, we further investigated the viability effect of compounds 6 and 7 in MDA-MB-435 cells, which were found to be somewhat more sensitive to the activity of both compounds (Tables 2 and 3), while they also possess more favorable features for the development of human-to-mouse xenografts [22,23].

 Of course, it is beyond any doubt that validating our findings in additional cell lines (resource constraints did not allow us to do this so far), with distinct genetic and phenotypic backgrounds, would strengthen the overall impact of the potential anticancer properties of these agents.

We also have addressed these limitations in the revised version of the manuscript. Please see the 'Discussion' section (lines 461-473) and the 'Conclusion' section (lines 477-483) for details. We acknowledge that future work will expand to multiple cell lines to confirm the generalizability of our results.

 Discussion, lines 466-478: A major limitation we must acknowledge in this study is the use of MDA-MB-435 as our sole model for studying the mechanisms of action. Firstly, its origin is debated [42], [43]. Secondly, it is not fully representative of melanoma. MDA-MB-435 is reported to be representative of amelanotic melanoma, which limits our ability to study the role of melanogenesis in melanoma progression and management, given that melanogenesis is re-ported to be closely linked to the advancement of melanotic melanoma [44].

Nevertheless, MDA-MB-435 cells remain an accepted model in several areas of tumour biology. For example, they are still used as a melanoma model in the NCI-60 panel and are employed by the COMPARE algorithm to draw significant conclusions [19]. Further-more, a wealth of information is available from the extensive analysis of the cell lines included in the NCI-60 panel while it has more favorable features over SK-MEL-28 for developing in vivo models [22], [23], which are necessary for further evaluation of the activity of these (adamantly) piperazine derivatives.

Conclusion, lines 482-488:  Among them, the fluorinated derivative 7 showed strong potency, highlighting the impact of the fluorination at the phenyl ring on biological activity. This study suggests that these compounds hold promise as potential therapeutic agents, especially against melanoma.  However, since these findings have only been observed in MDA-MB-435 melanoma cells, future studies must validate them in additional melanoma cell lines with different genetic and phenotypic characteristics to elucidate the precise molecular mechanisms involved.

5. SRB measures total protein, not viability. Clarify how it distinguishes cytostasis vs. cytotoxicity (e.g., via LC50 vs. GI50).

Authors’ reply:   We thank the reviewer for this important observation, and we have now clarified this in the Methods to ensure that readers understand how the SRB assay results were used to assess compound activity.

Materials and methods lines 137-144: Growth inhibition of 50% (GI50) is defined by the formula (Ti - Tz) / (C - Tz)) x 100 = 50. Total growth inhibition (TGI) is calculated by the formula Ti = Tz. Finally, the LC50, indicating a net loss of cells following treatment, is calculated using the formula (Ti-Tz)/Tz) x 100 = −50. In these formulas, Tz represents a measurement (absorbance) of the cell population for each cell line at the time of drug addition (i.e. 24 hours after plating the cells), C represents the measurement of the control cells and Ti represents the growth of cells in the presence of a given concentration of the drug, and at the end of the 48-hour incubation period [20].

  1. LC3-II accumulation alone doesn’t prove autophagy; include flux assays (e.g., bafilomycin A1) to confirm autophagosome turnover.

Authors’ reply: Thank you very much for your insightful comment on LC3 II accumulation and the importance of autophagic flux assays. We agree that LC3 II levels alone cannot definitively demonstrate the induction of autophagy, as these levels may reflect either increased formation of autophagosomes or impaired degradation of autophagosomes. Due to experimental constraints, we were unable to perform flux assays such as bafilomycin A1 or chloroquine treatment in this study. However, we have now revised the manuscript to clearly state this limitation, and to emphasise the need to interpret the LC3 II data with appropriate caution. We also discussed the need for future studies to incorporate flux measurements in order to conclusively assess autophagic turnover and flux. We believe that this contextualisation ensures that our findings are presented accurately and in the

Results, lines 325-326: These dynamic changes in the protein levels of LC3 A/B I and II suggest a potential involvement of autophagy in the observed cell death.

Discussion, lines 405-417: To explore the underlying mechanisms of cell death, we analyzed markers of apoptosis and autophagy in MDA-MB-435 cells treated with the compounds. Our current data demonstrate an increase in LC3 II levels, indicating the induction of autophagy. However, we acknowledge that this alone is insufficient to discriminate between increased autophagosome formation and impaired autophagic degradation. Moreover, the autophagic pathway may play a dual role: cell rescue initially, which is more evident at a concentration of 10 μM and with derivative 6, where cells can recover, as shown by the trypan blue results (Fig. 3a); and cell death at higher concentrations later on, probably due to irreversible damage caused by high concentrations of compound 6 and both concentrations of compound 7. Apparently, further studies addressing autophagic flux (autophagic degradation) with the use of inhibitors like bafilomycin A1 or chloroquine (which block lysosomal degradation) are required to confirm the full induction and the role of autophagy in the cell death mechanism of compounds 6 and 7.

Discussion, lines 439-443: The activation of this pathway was again found to be concentration- and time-dependent. The increased levels of LC3 II and the activation of apoptosis may indicate a pleiotropic mechanism of cell death for these compounds, which requires further clarification, especially regarding the role of autophagy, as previously mentioned.

  1. Caspase-3 cleavage is absent despite PARP data. Include caspase-3/9 assays to solidify apoptosis claims.

Authors’ reply: We thank the reviewer for this constructive comment. In response, we have included data on procaspase 3 levels in the revised manuscript, which now complements the PARP cleavage results. These findings provide further evidence of the involvement of apoptosis in the mechanism of action of the compounds. Although further strengthening of this conclusion would require caspase 9 assays, we acknowledge this as a limitation of the current study and have indicated that future work will expand to include analyses of upstream caspase activation.

Results lines 311-316: To further confirm the induction of apoptosis, we also evaluated the levels of pro-caspase 3. As can be seen in Figures 6a–e and 6b–f, treatment of cells with 20 μM of the compounds for 24 hours resulted in a substantial decrease in procaspase 3 levels, indicating the activation of caspase 3. Interestingly, this reduction in procaspase-3 levels appears to be more pronounced in the case of compound 7. These results are consistent with the results for PARP levels.

Discussion lines 441-448: In a subsequent study, combined evidence from caspase 3 and PARP cleavage supports the induction of apoptosis by both agents. The activation of this pathway was again found to be concentration- and time-dependent. The increased levels of LC3 II and the activation of apoptosis may indicate a pleiotropic mechanism of cell death for these compounds, which requires further clarification, especially regarding the role of autophagy, as previously mentioned. However, as with autophagy, future work needs to extend these findings by including upstream caspase assays to substantiate this pathway further and reveal the exact apoptotic pathway activated.

8. GBL is an atypical marker for mTOR. Use phospho-S6K/S6 or 4E-BP1 to directly assess mTOR activity.

Authors’ reply: Thank you for this valuable comment. We fully agree with the reviewer that GBL is not a standard marker for mTOR signaling, and we appreciate the recommendation to use phospho-S6K/S6 or 4E-BP1 as more direct indicators of pathway activity and explore potential interactions of our compounds with the mTOR complex at a structural/regulatory level. While we are unable to perform additional experiments at this stage due to time and resource constraints, we have revised the text to clarify the limitations of using GBL in this context and now explicitly acknowledge that canonical markers such as phospho-S6K/S6 or 4E-BP1 provide a more direct assessment of mTOR activity. We also explain that this main aim of studying GβL was to explore potential interactions of our compounds with the mTOR complex at a structural/regulatory level. We believe this clarification will strengthen the interpretation of our data and provide readers with appropriate context. Please see

Results lines 325-326: In light of the observed changes in LC3 II levels, we proceeded to evaluate the expression of GβL, a regulator of the mTOR pathway, in order to investigate the potential interaction of these compounds with the mTOR/PI3K/Akt axis at the structural/regulatory level.

Discussion lines 418-440: Given the central role of the mTOR/PI3K/Akt signaling axis in regulating autophagy and cancer in general [27], we examined the effects of these two compounds on the expression of the mTOR pathway regulator GβL/mLST8. This analysis aimed to primarily ex-plore potential interactions of our compounds with the mTOR complex at a structural/regulatory level and provide some preliminary insight into whether modulation of this pathway contributes to the anticancer effects observed with the compounds. GβL/mLST8 protein (G protein β subunit-like/mammalian lethal with SEC13 protein 8) is a core regu-latory subunit of both mTOR complexes, mTORC1 and mTORC2, that plays a pivotal role in mTOR kinase activity. In cultured human cancer cells, depletion of GβL/mLST8 has been shown to impair mTORC1 signaling [28-29], suggesting that GβL/mLST8 may have a role in cancer via modulating mTORC1/C2 complexes function. Although compounds 6 and 7 were not found to significantly alter GβL/mLST8 protein levels under the experi-mental conditions of this study, targeting GβL/mLST8 could be a promising approach to improving the effectiveness of mTOR inhibitors. In light of the data showing that modula-tion of impaired mTORC1 signaling can result in destabilized mTOR complexes, modula-tion of GβL/mLST8 might lead to impaired mTOR signaling, making cancer cells more susceptible to mTOR-targeted therapies by promoting cell death pathways such as apop-tosis or autophagy. Clearly, further studies investigating the combined effects of (ada-mantly) piperazines and mTOR inhibitors are needed to determine whether GβL contrib-utes to therapeutic responsiveness. Additionally, we must acknowledge that GβL is an in-direct marker of mTOR activity. Future studies using downstream effectors, such as phos-pho-S6K or 4E-BP1, are necessary in order to assess mTOR kinase activity directly in re-sponse to these compounds.

  1. Dose-Response Curves-Fig. 2/3 lack error bars or statistical significance markers. Add SEM and p-values for key comparisons.

Authors’ reply: Thank you for this valuable suggestion. In the revised versions of Figures 2 and 3, we have included error bars to represent the standard error of the mean (SEM). Regarding statistical significance, please note that p-values are reported for the calculated parameters (GIâ‚…â‚€, TGI and LCâ‚…â‚€), which are derived directly from these growth curves and described in the figure legends. We believe that presenting statistical significance in this way is more informative, as it allows readers to interpret the data in the context of the key pharmacological endpoints rather than individual data points.

10. Clonogenic Assay-Colony counts are qualitative (images) but quantified via SRB absorbance. Use direct colony counting for rigor.

Authors’ reply: Thank you for this helpful suggestion, which indeed strengthens the rigor of our work. We have now added colony counting to the analysis. We would also like to note that in our laboratory we routinely employ this assay in a dual role: first, to assess the effectiveness of the tested compounds against the most aggressive fraction of the cancer cell population (i.e., cells capable of surviving and proliferating under challenging conditions such as very low plating density), and second, as a long-term assay (please see M&M, line 187) to validate that the observed effects are time dependent and to define the lower activity limits of the studied agents.

Materials and methods lines 201-202: Colonies were also counted using ImageJ2 (Fiji version 1.54p).

Results: Figure 5.

11. Western Blot Quantification-Densitometry graphs (Figs. 6–7) lack normalization controls (e.g., actin for PARP, GBL). Reanalyze with proper loading controls.

Authors’ reply: We thank the reviewer for their careful consideration of our manuscript. However, we must point out that all values shown in the densitometric analyses were normalised against β-actin (please see the labelling on the y-axis in Figs 6 and 7). We, however, have revised the Y-axis labelling to clarify this (relative protein expression normalized to β-actin).

  1. Time-Course Gaps-Autophagy/apoptosis markers are shown at limited timepoints (e.g., 6–24h). Extend to 48h to align with viability assays.

Authors’ reply: We thank the reviewer for the careful insight into our manuscript. However, we must point out that the WB time points are aligned to the viability assays, especially the trypan blue assay, which was the actual guidance for further studies on the expression of these markers. Please see figure 4 which shows the Trypan blue results.

  1. The claim that fluorination enhances potency (p. 12) lacks mechanistic support. Compare logP/pKa or sigma receptor binding to justify.

Authors’ reply: Thank you very much for this important comment. We have calculated and now added a table on the ClogP and pKa values for our compounds, which showed a slightly higher lipophilicity for compound 7.  Furthermore, we have significantly expanded the discussion with references to major studies demonstrating how fluorination can modulate molecular properties of chemicals such as lipophilicity, metabolic stability, conformational flexibility, and even sigma receptor affinity which collectively influence drug bioavailability and target interactions.

Discussion lines 384-404: The two compounds 6 and 7 have similar pKa values, with compound 7 being slightly more lipophilic (ClogP = 4.58 vs. 4.43, Table 4).

Table 4. In-Silico calculated physicochemical parameters logP and pKa for compounds 6 and 7.

CLogP

pKa

Compound 6

4.434

8.956

Compound 7

4.577

8.950

 This modest increase, however, can enhance passive membrane permeability and facilitate greater intracellular accumulation, especially in the acidic tumor microenvironment, where weak bases undergo ion trapping. As a result, compound 7 achieves higher intracellular exposure, which plausibly explains its superior anticancer activity. Another significant difference between the two molecules is the presence of a fluoride atom in the phenyl group, which could explain the superior activity of compound 7. Fluorination is a strategy used in designing effective anticancer drugs, as fluorine atoms may enhance drug stability, lipophilicity, and cell permeability, and facilitate interactions with biological targets [25], even enhancing sigma receptor affinity [26]. Fluorinated compounds like 5-fluorouracil and approved drugs such as gefitinib (quinazoline, which is substituted by a (3-chloro-4-fluorophenyl) nitrilo group, a 3-(morpholin-4-yl) propoxy group and a methoxy group at positions 4,6 and 7, respectively) and capecitabine (cytidine in which the hydrogen at position 5 is replaced by fluorine and the amino group attached to position 4 is converted into its N-(penyloxy) carbonyl derivative) are used to treat various cancers. Nonetheless, direct mechanistic links need further detailed investigation.

  1. No experimental evidence ties effects to sigma receptors. At least cite overexpression in MDA-MB-435 (Ref. 11) or propose knockdown experiments.

Authors’ reply: We thank the reviewer for this valuable suggestion. We have added more references documenting the sigma-receptor relationship of adamantlyl derivatives, and we have expanded the discussion to highlight the potential involvement of sigma receptors in mediating the observed effects. We have also clearly stated that this mechanism requires direct confirmation through experimental validation using knockdown or overexpression approaches.

Discussion lines 449-461: Interestingly, adamantane analogues have been reported as antiproliferative and cytotoxic agents, while the cytotoxic activity was found to be related to their affinity for sig-ma-receptors [8], [15-17]. Sigma1 and sigma2 receptors have drawn attention to their po-tential therapeutic implications despite an incomplete understanding of their mechanisms of action. Sigma2 receptors are being explored as targets for cancer therapy and imaging agents and have been associated with the induction of autophagy [30] and apoptosis in cancer cells [31], [32]. Sigma1 has also been implicated in promoting cellular survival under oxidative stress by transcriptionally regulating Bcl-2 via the ROS-NF-κB pathway [33]. In addition, several studies have shown that sigma2 ligands can induce cell death in sev-eral cancer types through caspase-dependent and -independent apoptosis, lysosomal leakage, oxidative stress, Ca2+ release, ceramide production, autophagy, and cell cycle disruption [34-41]. Given the affinity of our compounds to sigma receptors, and the reported overexpression of these receptors in MDA-MB-435 cells [20], we hypothesize that the observed effects of these (adamantyl) piperazines may be mediated, at least in part, through sigma receptors. This hypothesis remains to be validated, and ongoing mechanis-tic studies of our group are directed toward elucidating these molecular interactions by using genetically modified cancer cells.

  1. The interplay is oversimplified. Discuss if autophagy is pro-survival (early) or pro-death (late), citing flux vs. apoptosis timing.

Authors’ reply: We thank the reviewer for this important comment. In light of the comments made by both reviewers and recognizing that the results based on LC3 II pattern expression changes are quite insufficient to conclude even for the full induction and function of autophagy by compounds 6 and 7 we have revised the manuscript by eliminating this part which is indeed oversimplified and pointing out the limitations of our study. Towards this aim we have revised the current title (please see response to comment 1), the abstract (please see response to comment 2) and results and discussion.

Results, lines 322-323: These dynamic changes in the protein levels of LC3 A/B I and II suggest a potential involvement of autophagy in the observed cell death.

Discussion, lines 405-417: To explore the underlying mechanisms of cell death, we analyzed markers of apoptosis and autophagy in MDA-MB-435 cells treated with the compounds. Our current data demonstrate an increase in LC3 II levels, indicating the induction of autophagy. However, we acknowledge that this alone is insufficient to discriminate between increased autophagosome formation and impaired autophagic degradation. Moreover, the autophagic pathway may play a dual role: cell rescue initially, which is more evident at a concentration of 10 μM and with derivative 6, where cells can recover, as shown by the trypan blue results (Fig. 3a); and cell death at higher concentrations later on, probably due to irreversible damage caused by high concentrations of compound 6 and both concentrations of compound 7. Apparently, further studies addressing autophagic flux (autophagic degradation) with the use of inhibitors like bafilomycin A1 or chloroquine (which block lysosomal degradation) are required to confirm the full induction and the role of autophagy in the cell death mechanism of compounds 6 and 7.

Discussion, lines 438-442: The activation of this pathway was again found to be concentration- and time-dependent. The increased levels of LC3 II and the activation of apoptosis may indicate a pleiotropic mechanism of cell death for these compounds, which requires further clarification, especially regarding the role of autophagy, as previously mentioned. However, as with autophagy, future work needs to extend these findings by including upstream caspase assays to substantiate this pathway further and reveal the exact apoptotic pathway activated

16. Figure 4: Label microscopy images with scale bars.

Authors’ reply: We thank the reviewer for the comment. We would like to note that a scale bar is already included in Figure 4 (please see Figure 4c). However, to be clearer we also have added in the caption legend the magnification which is the same for all images as the microscope used as well to take the images. This information has been added to the relevant section of trypan blue, results lines 274-275.

  1. Table 2/3: Highlight melanoma-specific results (bold/asterisks) for clarity.

Authors reply: We thank the reviewer for this suggestion. As recommended, we have highlighted the melanoma-specific results in Tables 2 and 3 using bold font to improve clarity and facilitate the quick identification of the relevant data in the revised manuscript.

  1. Typos: "pro-Caspase 3" "procaspase-3" (consistent capitalization).

Authors reply: We thank the reviewer for pointing out this typographical error. It has been corrected to “procaspase 3” throughout the manuscript to ensure consistent formatting and capitalization.

Round 2

Reviewer 1 Report

Comments and Suggestions for Authors

This reviewer thank the authors for the throughout revision of the manuscript.

The last paragraph of the introduction section ("focusing on their interaction 89 with sigma receptors", line 89) still sounds strange/irrelevant.

Otherwise the authors have addressed all my questions.

Please make sure the figures do not exceed page width in the final version (see figures 2).

Author Response

1.The last paragraph of the introduction section ("focusing on their interaction 89 with sigma receptors", line 89) still sounds strange/irrelevant.

Dear reviewer we deeply apologize for this oversight. Indeed this sentence makes no sense so we have replaced it with a more meaningful paragraph, which reads as follows:

Introduction lines 88-93: Building on our previous findings [18], the present study aimed to further characterize the anti-cancer activities of adamantyl-piperazine derivatives and elucidate aspects of their underlying mechanisms of action. Our results highlight the in vitro anti-cancer potential of these compounds, which displayed selective activity against melanoma under the examined experimental conditions. The data also suggest a pleiotropic mechanism of action regarding cell death, likely involving both apoptosis and autophagy.

2. Please make sure the figures do not exceed page width in the final version (see figures 2).

We also revised the figures and tables to align them with the text.

We hope that we have addressed your concerns properly.

Once again, we would like to express our deepest gratitude for the time and effort you have devoted to helping us strengthen our work.

Reviewer 2 Report

Comments and Suggestions for Authors

Manuscript is very fine after revision. Hence I accept this manuscript in current form

Author Response

Dear reviewer,
We would like to thank you once again for your kind words and for the time and effort you have devoted to helping us to substantially improve our work.